



# Effect of Blade Inclination Angle for Straight Bladed Vertical Axis Wind Turbines

Laurence Morgan[1], Abbas Kazemi Amiri[1], William Leithead[1], and James Carroll[1]

[1]EEE, University of Strathclyde, Glasgow, G1 1XW, Scotland

**Correspondence:** Laurence Morgan (laurence.morgan@strath.ac.uk)

**Abstract.** Vertical Axis Wind Turbines (VAWTs) have received renewed research interest in the offshore environment due to a number of design synergies that have the potential to decrease the cost of energy for offshore wind. Many studies have been completed on the rotor design for straight bladed (H) rotors however there is sparse information on the effect of blade inclination angle on VAWT aerodynamic performance, and the optimal blade design of VAWTs with inclined blades (V-rotors)
for maximum power capture.

This paper presents a systematic study into the effect of blade inclination angle, chord distribution, and blade length on VAWT performance. In the case of fixed chord length blades, it is found that significant power gains are available through blade inclination, between 10% and 68%, dependent on blade length. This is driven by the increase in rotor swept area. Further investigation indicates that despite this, under maximum blade stress limitations the most economical solution for fixed chord
length blades are H-rotors.

Optimal chord distributions to maximise the rotor power coefficient are then obtained, and a natural blade taper is observed. Significant power gains, between 10% and 69% dependent on blade length, are observed through blade inclination. However, consideration must be taken to limit blade mass. For a given power rating, whilst satisfying limitations on maximum blade root bending stress, it is found that blade volume can be reduced between 9% and 42% dependent on blade length, and rotor torque
can be reduced between 3% and 9%. This indicates the potential of V-rotors to reduce the cost of energy compared to H-rotors in traditional VAWT designs. Additionally, inclined blades are shown to increase the operational tip speed ratio, demonstrating their applicability to turbines using secondary rotors, such as the X-Rotor.

## 1  Introduction

### 1.1  background

Renewable energies are key to combating the ongoing climate crisis, and offshore wind energy is a integral pillar of this response (IRENA, 2022). It has be estimated that a three fold increase in the rate of deployment of wind energy is required by 2030 to meet our climate goals (Lee and Zhao, 2021) and decreasing the cost of energy for offshore wind can significantly increase the rate of offshore wind deployment. In this context Vertical axis wind turbines (VAWTs) have been identified as a technology that has the potential to significantly reduce the cost of offshore wind energy due to a number of design synergies





(Borg et al., 2012) (Sutherland et al., 2012). Further to this, recent work has demonstrated that a number of mechanisms allow for VAWT wakes to re-energies significantly faster than those of traditional horizontal axis wind turbines (HAWTs), facilitating increased wind farm density (Huang, 2023) and further reductions in the cost of energy. This has lead to significant academic and commercial interest in the design concept (Hand and Cashman, 2020).

VAWTs have many potential configurations, with any 2D shape with vertical symmetry representing a possible rotor con-
figuration. For large offshore structures, practical design considerations have led to a convergence on 2 key designs: Darrieus VAWTs, with curved blades attached at both the blade root and tip, and straight bladed VAWTs, typically configured with blades orientated parallel to the axis of rotation in an H configuration. Straight bladed designs with inclined blades have also been proposed (Ljungström et al., 1974; Musgrove, 1977; Sharpe and Taylor, 1983; Shires, 2013a; Leithead et al., 2019), these rotors will be henceforth referred to as V-rotors.

The use of V-rotors was first introduced by Ljungström (Ljungström et al., 1974), initially referred to as 'Y' rotors. It was noted in this work that straight bladed VAWTs facilitated cyclic pitching more easily than Darrieus type. This rotor concept was also briefly described by Park (Park, 1976). A variable geometry H-Rotor with inclinable blades was later proposed by Musgrove (Musgrove, 1977), this used a reefing system to incline the blades as a means of reducing power capture at high wind speeds. Wind tunnel tests on a scale model conducted by Willmer (Willmer, 1980) demonstrated that the power
coefficient decreased significantly as the blade inclination angle was increased from 0° to 60°. Figure 1a) shows the 'Musgrove 250' turbine with blades in the 'reefed' position.

David Sharpe further developed Ljungström's rotor concept from 1983 onwards (Sharpe and Taylor, 1983). These designs did not utilise a cross-arm, with the blades attached directly to the rotor hub. Listed benefits of the design included: The reduction in capital costs due to the short rotor tower and the lack of cross-arm, the ease of manufacturing of straight (untwisted)
blades, the potential for control with either variable geometry or variable pitch, and the increased portion of the rotor swept area sampling higher wind speeds at high altitudes. An initial study determined the power coefficient for inclinations angles of 35°, 45° and 55° for an blade aspect ratio of 11.9, finding the maximum power coefficient occurring with a inclination angle of 55°. Additionally, increasing blade aspect ratio was found to increase the peak power coefficient for an inclination angle of 45° (Sharpe and Taylor, 1983). The aerodynamic simulation work was completed using a BEM based double multiple streamtube
theory (DMS) approach described in (Sharpe, 1984). Further work was completed, validating the aerodynamic simulation work and demonstrating how the rotor could be controlled using a tip-pitch system in (Robotham et al., 1985), and a 5kW system was field tested in 1987 (Sharpe et al., 1987), this system is shown in figure 1b). Following these tests, a single bladed design utilizing a counterbalancing weight was proposed (Shires, 2013a), this design is shown in figure 1c). Following this period, the development of V-Rotor concepts was predominantly undertaken by private engineering companies. Altechnica continued to
work on the concept through the 'Taylor V-Turbine' as shown in figure 1d).

The next significant academic work on V-rotors was undertaken in the NOVA project with in the design of the Aerogenerator-X as shown in figure 1e) (Shires, 2013a). This work was conducted as partnership of a number of British academic institutions in conjunction with Wind Power ltd. Here, a 10MW V-rotor was designed, the rotor blades were augmented with sails, aimed and counteracting the aerodynamic overturning moments inherent in V-VAWT designs, the rotor design work is presented in

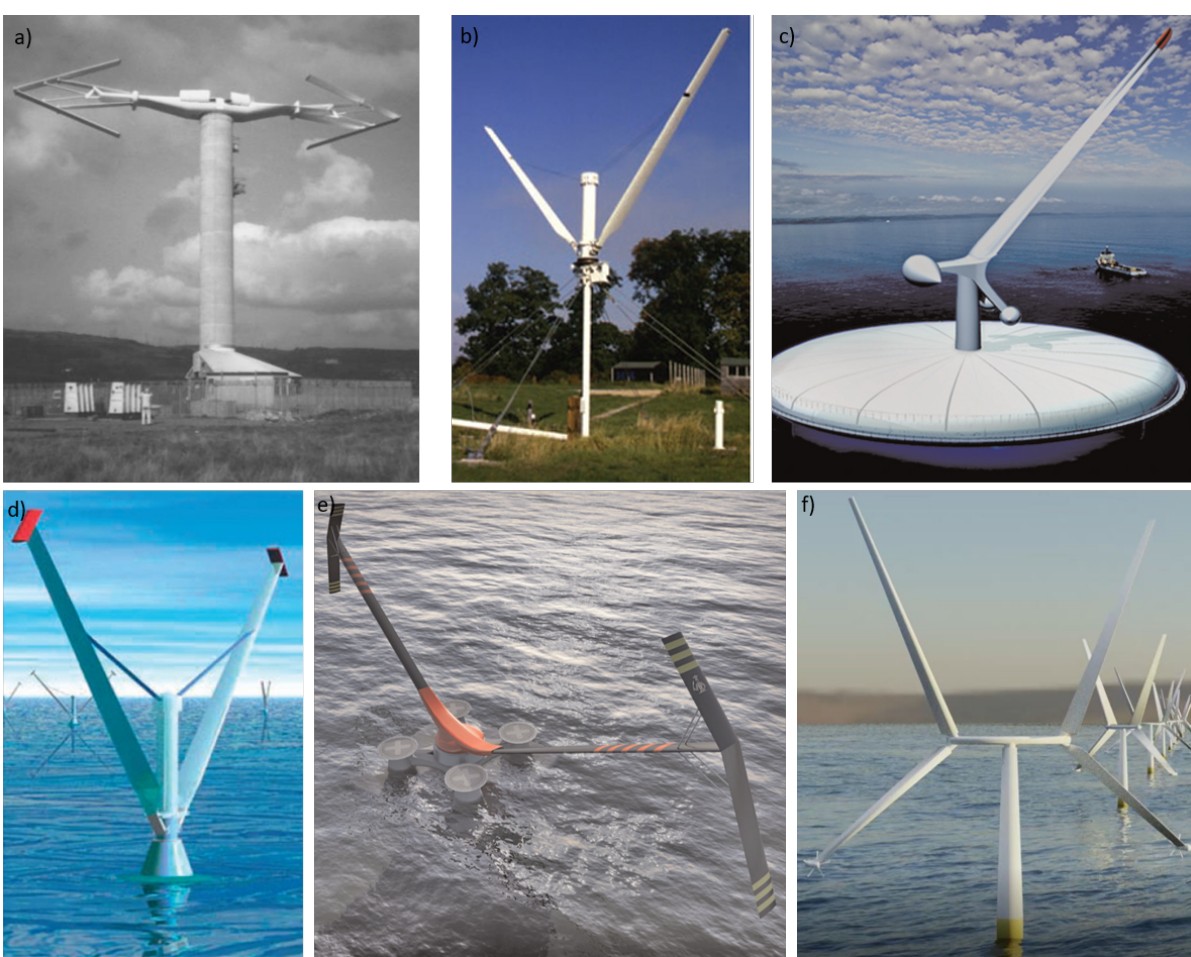

**Figure 1.** a) Musgrove 250 turbine (Price, 2006), b) 5kW V-Rotor tested by Sharpe and Taylor (The Open University, 2015), c) Single bladed V-VAWT concept (Shires, 2013a), d) Taylor V-Turbine (The Open University, 2015), e) The aerogenerator concept (Shires, 2013a), and f) the X-Rotor wind turbine concept.





(Shires, 2013a). A gradient based optimisation approach was coupled to a DMS based aerodynamic solver (Shires, 2013b) and
was used to find a rotor geometry that minimised the rotor volume (used as a proxy for the rotor cost) under a number of design
constraints including ensuring a rated power of 10MW at 13m/s with rated rotor speed of 4rpm, the limitation of the maximum
overturning moment and a constraint on the maximum length of unsupported sails and struts. The completed optimisation was
found to decrease the rotor volume by 3% whilst preserving the aerodynamic power and decreasing the overturning moment by

11%. For some of the optimisations presented, the inclination of the rotor blades was a free variable, however the optimisation
procedure did not significantly alter the initial inclination angle of $59°$.

Following the completion of the NOVA project, the X-Rotor wind turbine concept began development at the university of
Strathclyde (Leithead et al., 2019) and further research is underway in the form on an EU H2020 project (Cordis, 2023). The
turbine consists of an X-shaped primary rotor, made up of an upper and lower V, which utilises secondary rotors, attached

to the tips of the lower V, for power take off. The use of inclined rotor blades is critical in the X-Rotor concept as a means
of increasing the primary rotor tip speed ratio, which is key to ensuring efficient power conversion between the primary and
secondary rotors Morgan et al. (2024). A rendering of the turbine concept is shown in figure 1e). The primary rotor blades are
attached to a central cross-arm with a radius of 25m and extend to a tip radius of 75m, with inclination angles of $30°$ and $130°$
respectively. The lower blades reduce the overturning moment in a similar manner to the sails of the aerogenerator concept,

and provide an attachment location for the secondary rotors. The aerodynamic and structural design of the rotor is presented
in (Leithead et al., 2019), however the aerodynamic design process of the primary rotor was not presented. A DMS model
was developed and validated against higher fidelity lifting line free vortex wake simulations to simulate the X-Rotor turbine
in (Morgan and Leithead, 2022) and comparison of multiple aerodynamic codes simulating the primary rotor is presented in
Giri Ajay et al. (2023).

Absent from the current work published on V-rotors, as represented in this literature review, is a systematic study on the effect
of key rotor design parameters, particularly the blade inclination angle, on aerodynamic performance. This hinders the design
of V-rotors at the conceptual stage. There is some sparse data on the effect on rotor inclination angle for specific blade/rotor
designs at the lab scale (Willmer, 1980; Sharpe and Taylor, 1983), however these show conflicting trends. Additionally, whilst
the effect of solidity and aspect ratio have been well studied for H-rotors, thus far these parameters have not been significantly

investigated for V-rotors. In the case of rotor solidity, this may be critically important as V-rotor blade sections operate at
variable tip speed ratios, indicating that blades with a variable chord distribution are likely to increase aerodynamic efficiency.

Additionally, the current literature on V-rotors does not explicitly describe the effect of rotor geometry on power capture.
Results are typically presented with respect to the rotor power coefficient $C_P$, however it is unclear whether the normalisation
area is defined with the initial turbine area, or the rotor with inclined blades. A key benefit of the V-rotor concept is the increase

in swept area achievable by inclining a blade of fixed length, this must be explicitly accounted for when presenting results.

## 1.2   Research contribution

This paper presents the results from a systematic numerical study of straight bladed VAWT configurations, completed to
understand the effect of blade inclination angle, rotor aspect ratio, and blade chord length on rotor aerodynamic performance





and investigates optimal blade design for V-rotors. Firstly, fixed chord length blades are considered and the effect of chord length, inclination angle, and rotor aspect ratio are investigated in terms of power production, thrust loading, and optimal tip speed ratio. Following this, the data from these simulations are used to define plan-form blade designs that are optimised for maximum power capture. The effect of blade inclination angle and design tip speed ratio on the optimal blade plan-form are then discussed. Finally, the aerodynamic performance of these optimal blades is presented, and the opportunity for cost of energy reduction is discussed.

A systematic approach based on a grid search, rather than an optimisation tool, is used as it allows the wider design space to be studied and understood, allowing for broad trends to be identified, and engineering judgements to be applied without the formulation of specific cost functions. The investigation will use a single aerofoil section (NACA0018) and will not consider twisted or pitched rotor blades, as the introduction of these design variables would considerably increase the design space, these limitations are discussed in section 2.3.

In conclusion this paper provides:

1. A systematic study into the effect of blade inclination angle and rotor aspect ratio on the optimal design and aerodynamic performance of straight bladed VAWTs.

2. A study into the optimal chord distribution of inclined VAWT blades.

3. An evaluation of the potential for inclined blades to lower blade volume and rotor torque.

## 1.3 Paper structure

Section 2 describes the methodology used for numerical simulation, rotor geometry generation and blade geometry generation. Section 3 presents the key results from the numerical study, including the rotor performance parameters and optimum blade design, and section 4 discusses these results. Finally section 5 provides an overview of the key research outcomes and discusses avenues where work could be continued.

## 2 Methodology

### 2.1 Numerical Simulation

The numerical simulation of VAWTs is considerably more complex than that of HAWTs, both at the rotor scale and at the blade element scale. At the rotor scale, the flow interacts with the rotor surface twice, initially in the upwind, then in the downwind sweep. At the blade element scale, large cyclic variations in the angle of attack experienced along the blade which further complicate numerical simulation. This work utilises a 2 Dimensional Actuator Cylinder (2DAC) approach for the numerical simulation of the rotor aerodynamics.

The 2DAC model, first developed by Madsen (Madsen, 1988), discretises the rotor along it's vertical axis into 2D rotor segments. These circular rotor segments are treated as 2D actuator surfaces, and the induced velocity field is obtained through




solving the Euler equations in two dimensions. The loading of the actuator surface is coupled to the integrated loads on the
blade element, which are calculated using tabulated polars. These are coupled back to the solution of the flow field as the blade
loads are dependent on the both the relative velocity and angle of attack at the blade element. The 2DAC model used in this
study was developed by TUDelft as part of the X-Rotor project (Ferreira, 2021), a linearised solution to the Euler equations
is employed, as presented in (Madsen et al., 2013), and a Prandtl tip loss function is applied (Prandtl et al., 1927). Due to the
large number of simulations required, simulations were completed with a course discretisation of 20 blade elements per rotor,
with an angular step of $5°$.

The 2DAC model was selected as it is momentum based, and can therefore complete simulations in significantly less com-
putational time than higher fidelity approaches such as free vortex wake models and viscous computational fluid dynamics.
This was important due to the large number of simulations required for this parametric study. Whilst DMS methods are also
computationally light, it has been shown that 2DAC models more accurately reproduce the results of higher fidelity models
over a range of rotor design parameters and operating conditions (Ferreira et al., 2014). Specifically 2DAC modelling has
recently been shown to accurately reproduce rotor averaged performance parameters for inclined rotors in the case where pitch
offsets remain small (Giri Ajay et al., 2023). There are, however, many ways in which this model does not directly reflect the
behaviour of an actual rotor installed in realistic environmental conditions. These limitations in the modelling methodology are
listen below:

1. Dynamic stall modelling: The delay of flow separation around a blade section as the angle of attack exceeds the static stall
   angle, and the corresponding increase in the lift coefficient beyond the maximum static value is referred to as dynamic
   stall. This effect is present during VAWT operation (Laneville and Vittecoq, 1986), and has been shown to considerably
   alter VAWT rotor performance (Marten et al., 2016; Bianchini et al., 2018). As the blade root of a V-VAWT is operating
   at a low local tip-speed ratio and often with a large chord to radius ratio, the flow conditions are often out-with the range
   of validity that engineering based dynamic stalls models have been validated, i.e. at a local tip speed ratio bellow 1.
   Dynamic stall corrections have therefore not been included in order to limit the uncertainty in model prediction, and to
   provide transparency in the results.

2. Flow curvature modelling: Flow curvature effects have been shown to be significant for VAWT aerodynamic behaviour
   (Rainbird et al., 2015), however have been ignored in this investigation. This study is attempting to provide a general view
   of the effect of rotor geometry, rather than provide a highly accurate solution to the aerodynamic behaviour of a rotor
   with a specific geometry. A single aerofoil section (NACA0018) represented by a single set of polars, uncorrected for
   flow curvature, are used throughout this study. It could be considered that the geometrical blade sections have undergone
   an inverse conformal mapping procedure as in (Rainbird et al., 2015), or that a twist distribution has been included that
   accounts for the effective incidence angle as demonstrated in (Bianchini et al., 2016).

3. Modelling of 3D effects: Due to inclined rotor blades, induced velocities in the vertical direction that impact the flow
   conditions at each blade section will be present. In addition to these, radial flows will also be present and are expected
   to modify the blade element characteristics. Both of these effects are inherently ignored in the 2DAC model. However,





quasi-2D models have successfully been used to study rotors with inclined blades (Shires, 2013b), and have been shown
to successfully reproduce the aerodynamic characteristics of 3D models for rotor with inclined blades (Morgan and
160 Leithead, 2022).

4. Inclusion of Wind shear: This study has ignored the effects of wind shear both on power capture and on rotor design in
order to preserve the generality of the results. Appendix A discusses this issue further. Any detailed rotor design process
should include the effects of wind shear.

5. Strut modelling: The aerodynamic drag arising from strutting has been shown to reduce rotor power performance for
VAWTs (Worstell, 1981), however in this study the rotor support structure has not been considered. There is a wide
range of strut configurations proposed for different V-rotor concepts, as shown in figure 1, and for this reason only the
aerodynamically active parts of the rotor have been considered in this analysis.

In order to to provide a realistic optimal aerodynamic design for an inclined rotor, each of these limitations must be addressed
in the modelling. However, this paper is not attempting to find a single optimum rotor design, but is providing a survey of the
170 design space in order to inform readers on the trends associated with increasing rotor inclination angle, and to demonstrate the
potential benefits in power production available to rotors with inclined blades.

## 2.2 Rotor geometry generation

In this paper, the rotor geometry refers to the shape of the rotor swept area, independent of the blade geometry. As the rotors
considered in this study are straight bladed, the rotor geometry can be defined by the inner radius $r_0$, the blade length $l$, and the
175 inclination angle $\psi$. For generality, the blade length can be non-dimensionalised by the inner radius, $l_r = l/r_0$; this is equal to
double the rotor aspect ratio for an H-rotor. For this study rotor geometries are generated with $1 \leq l_r \leq 4$, and $0° \leq \psi \leq 50°$.
As mentioned in section 2.1, only the aerodynamically active sections of the rotor are considered here, and neither the tower
nor rotor struts are modelled. The variables $l, r_0$ and $\psi$ are labelled in figure 2 for clarity. For a given rotor configuration, the
frontal area can be calculated with

180 $$A = r_0^2 \left[ 2l_r \cos(\psi) + l_r^2 \cos(\psi) \sin(\psi) \right].$$ (1)

## 2.3 Blade plan-form generation

The blade designs considered within this study are split into two categories, the first are fixed chord length blades, and the
second are blades with an optimised chord distribution. The chord length is presented as non-dimensionalised by the inner
radius of the rotor, and the chord lengths considered in this study ensure that the maximum solidity at the blade root does not
exceed 1, i.e.

$$\sigma = \frac{Nc}{r_0} cos(\psi) \leq 1.$$ (2)





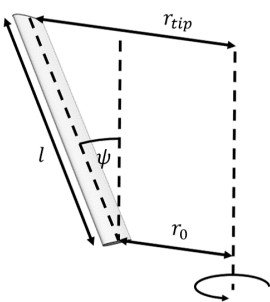

**Figure 2.** Geometry of a V-rotor.

where $N$ and $c$ represent the number of blades and chord length, respectively. For a two bladed rotor, the maximum non-dimensional chord length ranges from 0.5 for $\psi = 0°$ up to 0.775 for $\psi = 50°$. The full range of non-dimensionalised chord lengths considered for this study is given in table 1.

In order to find the optimum chord distribution, the the 2D rotor segments are assumed independent, an assumption inherent within the 2DAC approach, and the optimum blade plan-form shape is generated through finding the chord length that locally maximises the power coefficient for each rotor segment, using data from the fixed chord length simulations.

This approach can be repeated over a range of tip speed ratios, $1.5 \leq \lambda' \leq 6$, defining a set of blades optimised over a range of tip speed ratios. The optimum blade design for power capture can then be readily identified as the iteration that maximises $C_P$. The full range of solutions can also be explored to find potentially more practicable solutions and to identify tends within the design space. This approach can be repeated over the full range of rotor geometries, giving a set of optimal blade shapes designed for maximum $C_P$ over a 3 dimensional design space, covering $\lambda', \psi, l_r$. The full details of the completed simulations are given in table 1, in total 7332 rotor simulations were completed.

In this study the NACA0018 aerofoil section was used over the full blade length, with blade polars were generated at a Reynolds number (15 million) using X-Foil (Drela, 1989) and extrapolated using the Viterna method (Viterna and Janetzke, 1982). Whilst the optimal blade plan-form is dependent on the blade section used over the blade, this study is attempting to understand the general trends in rotor behaviour based on the rotor geometry, rather than provide a detailed optimal design case. It is expected that trends in rotor design will be consistent over a range of symmetric aerofoils, therefore the use of a single aerofoil section is not considered a significant limitation. The NACA 0018 section was selected as it is commonly used for VAWT rotors, and the Reynolds number was selected to be consistent with multi-MW offshore VAWT turbine concepts such as the X-Rotor.

The effect of blade number is also absent from this investigation. As this study focuses on the rotor averaged performance parameters and utilises an aerodynamic model based on revolution averaged loads, the blade number has only 2 effects: firstly to modify the rotor solidity, and secondly to reduce the tip losses; marginally increasing both power production and rotor loads. The dominant effect is expected to be the change in rotor solidity, which can be readily understood through an equivalent change in the blade chord through equation 2. Whilst the effect of blade number is not directly studied, and the effect of blade





| $\psi$ [°] | $l_r$ | $c/r_0$ | $\lambda'$ |
|---|---|---|---|
| 0 | [1,2,3,4] | [0.025,0.05,0.075...0.5] | [1.5,2,2.5,2.75,3,3.25,3.5,3.75,4,4.5,5,5.5,6] |
| 10 | [1,2,3,4] | [0.025,0.05,0.075...0.5] | [1.5,2,2.5,2.75,3,3.25,3.5,3.75,4,4.5,5,5.5,6] |
| 20 | [1,2,3,4] | [0.025,0.05,0.075...0.525] | [1.5,2,2.5,2.75,3,3.25,3.5,3.75,4,4.5,5,5.5,6] |
| 30 | [1,2,3,4] | [0.025,0.05,0.075...0.575] | [1.5,2,2.5,2.75,3,3.25,3.5,3.75,4,4.5,5,5.5,6] |
| 40 | [1,2,3,4] | [0.025,0.05,0.075...0.65] | [1.5,2,2.5,2.75,3,3.25,3.5,3.75,4,4.5,5,5.5,6] |
| 50 | [1,2,3,4] | [0.025,0.05,0.075...0.775] | [1.5,2,2.5,2.75,3,3.25,3.5,3.75,4,4.5,5,5.5,6] |

**Table 1.** List of rotor design and operational variables permuted over to generate data for this study.

number cannot be directly obtained from the existing data due to the changes in tip losses, the trends in rotor behaviour will be reproduced in terms of solidity, and this is not considered a significant limitation to the study.

## 3 Results

Section 3.1 presents the results of studying the effect of rotor inclination angle on fixed chord length blades, examining the relationship between blade inclination angle and rotor averaged variables, including the thrust, power, and optimal tip speed ratio. Variables pertinent to the cost of energy are introduced, and limitations on the maximum bending stress in the blade are investigated.

Following this, the blade optimisation procedure described in section 2 is demonstrated, and the resultant optimum blade plan-form shapes are discussed in section 3.2.

Section 3.3 evaluates the aerodynamic behaviour of rotor designs utilising the optimised blades presented in section 3.2. Following this, variables influential on the cost of energy are discussed, and limitations on maximum bending stresses are once again imposed to evaluate the practicality of rotor designs.

In order to understand how the rotor geometry effects rotor design, two variables are introduced here, the power product, and the thrust product. They are given by the relevant rotor performance coefficients scaled by the rotor area, to demonstrate aerodynamic and geometrical effects of rotor design:

$$\Phi_P = C_P A \tag{3}$$

$$\Phi_T = C_T A \tag{4}$$

where $A$ represents the rotor area. When normalised with respect to the values obtained for an equivalent H-Rotor, these variables allow the direct comparison between the power capture and rotor loads for rotors of different swept area.





## 3.1 Survey of design space fixed chord

Figure 3 displays the maximum power product, the corresponding thrust product, and the optimal tip speed ratio at which the
maximum power coefficient is reached as a function of the rotor inclination angle and the chord length of the blade. For ease
of comparison, the power and thrust products are normalised by the values obtained for the optimal H-Rotor. The blue square
marker represents the location of the optimal H-Rotor design, whilst the red diamond marker indicates the largest achievable
power product. The grey shaded area represents the region which has not been simulated, due to the restrictions on chord length
discussed in section 2. This is repeated over the range of aspect ratios introduced in section 2.

The rotors were simulated up to a maximum tip speed ratio of 6. It is clear that with the large inclination angles and high
blade lengths, the optimum power coefficient is not reached for the low chord length cases. This implies that the decrease in
maximum power product for these cases is artificially smaller due to the limited search range, and that a high power product
may be achievable.

Figure 4 (a) presents the maximum power product for the optimal straight H-Rotor blade ($c = 0.2r_0$) as a function of rotor
inclination angle for a range of rotor aspect ratios alongside the rotor area. Both variables are normalised by the respective
values for $\psi = 0$. Figure 4 (b) presents the corresponding tip speed ratio at which these maximum variables are achieved. From
this figure it is again clear that, for combinations of high aspect ratios and high inclination angles, the maximum tip speed ratio
range considered in this study is not sufficiently large to capture the maximum achievable power coefficient. Cases where the
optimal tip speed ratio is limited by the design range used within the study are highlighted in red.

Whilst the potential for power gains are clear from figures 3 and 4, it is important to understand the effects of any rotor design
on the cost of energy. At an early design stage, blade cost can be assumed to scale with blade mass, and whilst a detailed design
structural design is required to obtain blade mass estimates, the blade volume can be used as a proxy variable to understand the
relative change in blade mass between different designs. This is given by

$$V = l \int_0^1 k_1 c(x)^2 dx, \tag{5}$$

where $k_1$ represents a shape parameter, and $x$ represents the non-dimensional spanwise coordinate. In this study, the shape
parameter is constant along the blade as a single aerofoil section is used. In addition to blade mass changes, variations in the
rotor geometry, specifically the tip radius $r_{tip}$, and design tip speed ratio effect speed/torque input to the drivetrain. Assuming
similarity scaling, the drivetrain torque can be used as a proxy for the generator mass and cost. The rotor torque is given by:

$$Q = \frac{P}{\Omega} \tag{6}$$

with the rotor speed, $\Omega$, given by

$$\Omega = \frac{\lambda' U_0}{r_{tip}}, \tag{7}$$

and the tip radius given by

$$r_{tip} = r_0^2 \left[1 + l_r \sin(\psi)\right]. \tag{8}$$

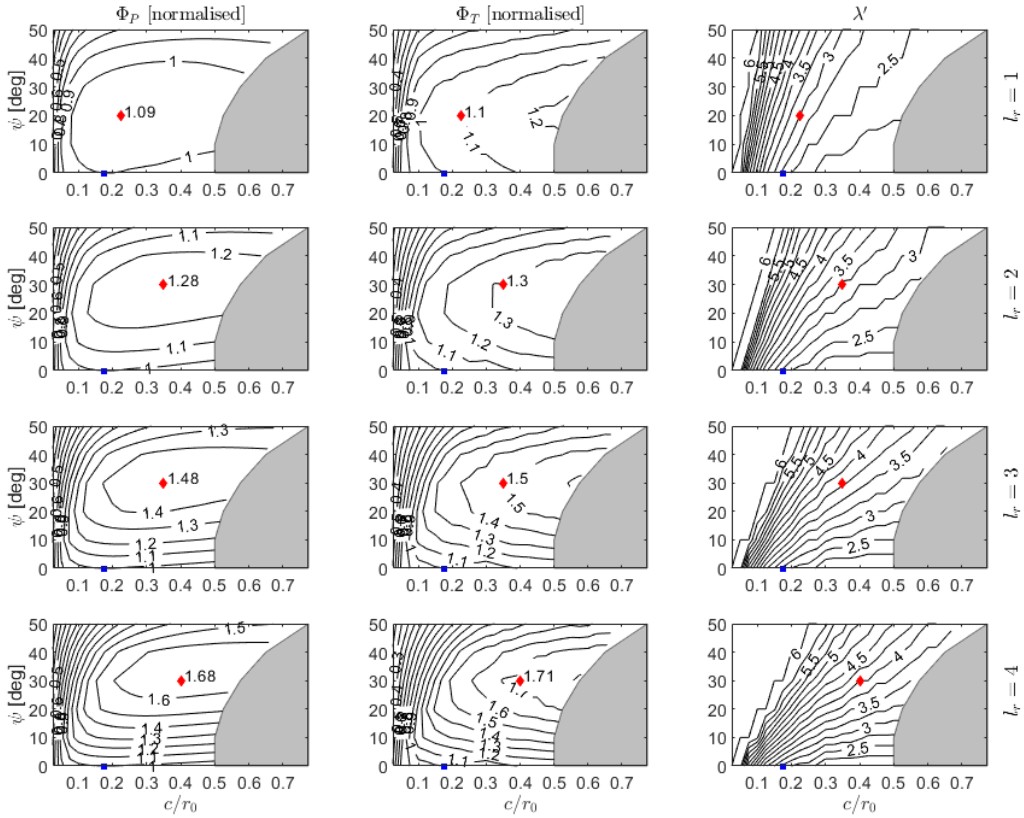

**Figure 3.** Power product, thrust product and optimal tip speed ratio, for untapered VAWT designs, presented as a function of $c/r_0$ and $\psi$, with thrust and power normalised by the optimal H-Rotor design. Results shown for blade lengths of $l_r = [1,2,3,4]$. The optimal H-rotor design is indicated by a blue square, and the optimal V-rotor design is indicated by a red diamond.

In order to compare values over different rotor designs, it is instructive to compare rotors with equal power production, this can be achieved through introducing the linear scaling parameter $s$, with

$$\Phi_P^{[1]} = s^2 \Phi_P^{[2]}. \tag{9}$$

The scaling parameter defines how the length scale of rotor design [2] must be modified in order to equal the power rating of design [1]. The inner radius, blade length, blade chord scale linearly with $s$, the change in rotor swept area, blade volume, and rotor torque then scale with equations 1, 5, 6 respectively.

A final consideration is the bending stresses within the blade. A blade with a specific material design will be capable on withstanding a certain maximum bending stress. In comparing blade designs, it is important to compare designs that have a similar maximum bending stress, otherwise material innovations are implicitly included within the comparison. Treating the





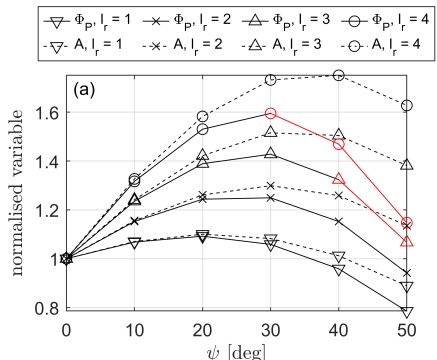 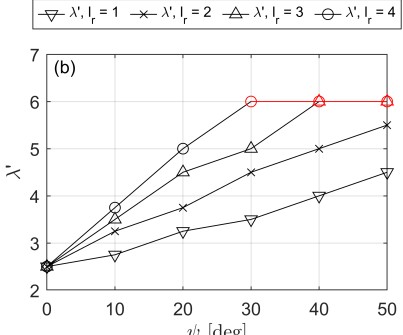

**Figure 4.** (a) Rotor power and rotor area for a blade of $c_r = 0.2$ normalised by the H-rotor values presented as a function of inclination angle. (b) Corresponding optimal tip speed ratio. Values where trends indicate the optimal tip speed ratio extends beyond $\lambda' = 6$ are marked in red.

blade a simple beam, the out of plane bending stresses at span-wise location $x$, and a distance $\delta$ from the structural center of the blade (normalised by the chord length $c$) are given by

$$\sigma(x) = \frac{\delta c(x)}{k' t(x)^3 c(x)} L \int_x^1 F_N(\tilde{x}) \tilde{x} d\tilde{x} = \frac{\delta}{k'' c(x)^3} L \int_x^1 F_N(\tilde{x}) \tilde{x} d\tilde{x}, \tag{10}$$

where $k'' \equiv k' \left(\frac{t}{c}\right)^3$ represents a shape function depended on the thickness of the aerofoil section, as a constant aerofoil section is used, $k''$ remains constant. As the blade has a fixed chord length, it is clear that the maximum bending stresses occur at the blade root. For the comparison presented here, the spanwise force distribution that maximises the bending stress at the design tip speed ratio is selected.

Figure 5 displays the blade volume and rotor torque for rotor designs of equal rated power as function of inclination angle and blade chord length. The values are normalised by the design values for the optimal H-Rotor. Alongside these contours, the maximum operational bending stresses are presented, normalised by the maximum bending stress for the optimised H-rotor. The constant blade volume contours have a positive gradient up to $20°$ for the $l_r = 1$ and up to $30°$ for $1 \le l_r \le 4$, indicating that rotors utilising inclined blades can reduce the cost of blades for rotors of the same power output. However, the maximum

bending stress contours clearly indicate that inclined rotor blades are subject to larger aerodynamic bending stresses. In all cases, the gradient of the maximum bending stress contours are larger than the gradient of the blade volume contours, this implies that by moving along each constant stress contour the minimum blade volume and rotor torque is obtained at an inclination angle of $0°$. This results implies that any material innovation that facilitates a blade design which can withstand larger aerodynamic bending stresses will be best applied to lower the chord length of the H-Rotor design if the objective is to

minimise blade costs.

Similarly, the contours in of constant maximum bending stress have a larger gradient than the rotor torque contours, again indicating the most optimal solution to reduce rotor torque, for a given blade material layup, is the H-rotor design. These



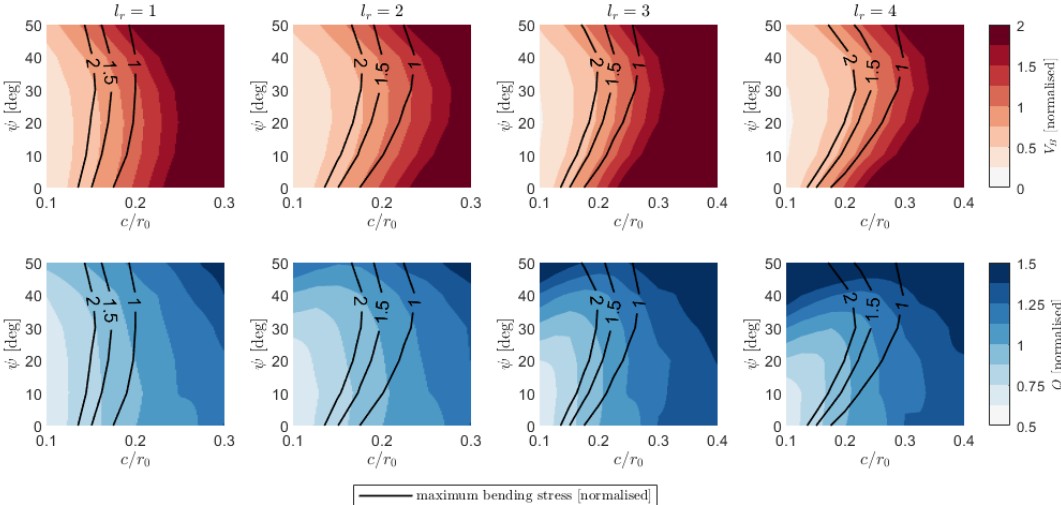

**Figure 5.** Blade volume and rotor torque for rotors of equal rated power, normalised by the optimal H-rotor design, presented as a function of chord length and inclination angle for $l_r = [1, 2, 3, 4]$. Contours of maximum bending stress, normalised by the value for the optimal H-rotor are presented alongside in black.

results indicate that; for straight bladed, fixed chord length VAWTs, inclined rotor blades do not represent an effective design innovation to lower the cost of rotor blades or drive-trains if blade design is driven by operational loads.

## 3.2 Blade Optimisation

The blade of a V-rotor experiences a varying local tip speed ratio and, at each rotor section, the chord length is normalised by a different radius to obtain the local solidity. This indicates that the optimal plan-form shape of a V-Rotor blade may not be straight. The process for generating an optimal plan-form blade shape is described in section 2. Figure 6 demonstrates the blade optimisation procedure at tip speed ratios from 2.5 to 4.5, for inclination angles of $0°$ to $50°$ and $l_r = 2$. The solid black line represents the chord length that maximises the local power coefficient at each span-wise location. There are a number of trends that are well demonstrated by Figure 6.

Firstly, the optimal chord length decreases as the design tip speed ratio increases, this has already been observed in figure 3 where the optimal tip speed ratio is seen to decrease with increasing chord length, as is standard for VAWTs (Hand et al., 2021). As the thrust coefficient is dependent on the product of the tip speed ratio and the rotor solidity, to reach optimal loading any increase in tip speed ratio should be matched by a decrease in solidity. Secondly, the optimal chord length is found to increase with increasing inclination angle. This can be understood as the chord length having to increase with $1/\cos(\psi)$ to preserve the rotor solidity and therefore optimal loading.

It is also clear that the introduction of an inclination angle promotes a tapered blade. For basic momentum methods it can be shown that, if linear lift coefficient is assumed, the rotor loading is directly proportional the the product of solidity and tip speed

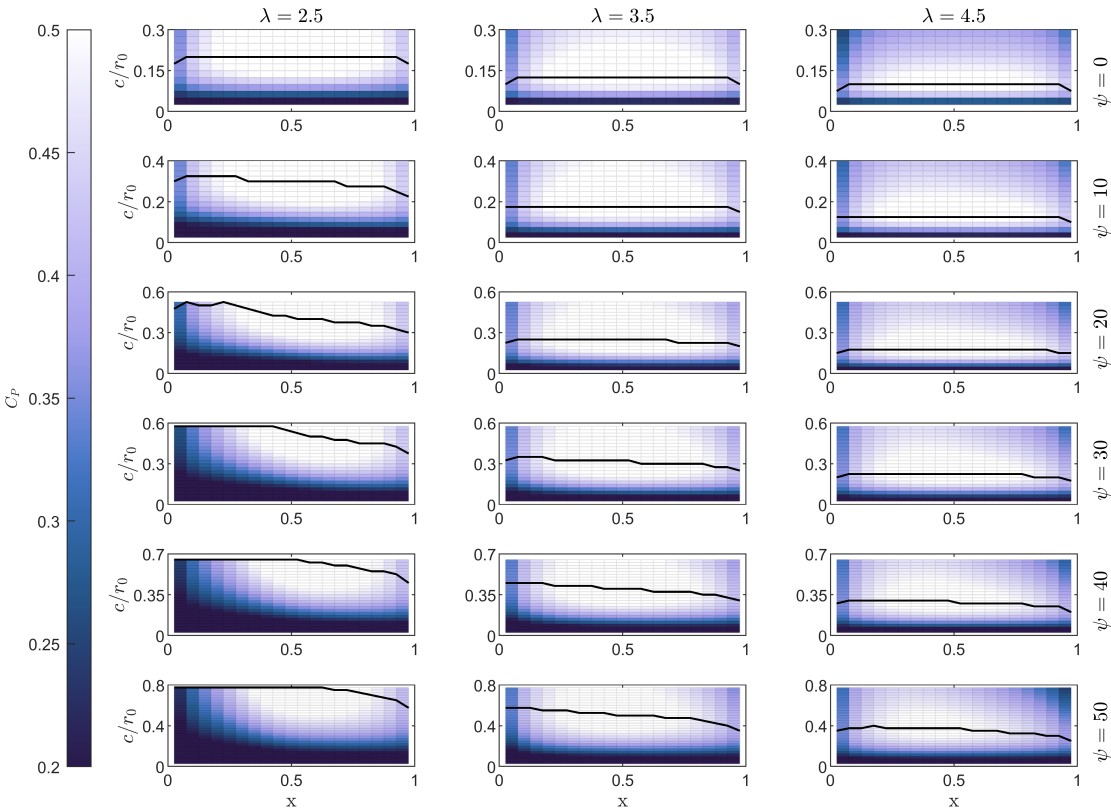

**Figure 6.** Optimal blade plan-form displayed above a power coefficient heat-map, for $l_r = 2$. Designs cover $\lambda' = [2.5, 3.5, 4.5]$ and $\psi = [0, 10, 20, 30, 40, 50]°$. Not the change in yaxis scale at different inclination angles due to the variations in maximum chord length in reference to equation 2.

ratio (Loth and McCoy, 1983). This would imply that the optimal inclined blade has a constant chord length. In reality however, as the rotor inclination angle increases, the local tip speed ratio decreases moving inboard from the tip. At low local tip speed ratios, the blade sections undergo large fluctuations in angle of attack, outside of the range in which the linear assumption is valid. At large angles of attack, the blade stalls and the lift coefficient drops significantly, this means that, to ensure the rotor is loaded more optimally, the chord length must increase.

Figure 6 also demonstrates a limitation of the current study. The limitation imposed by equation 2 to keep the local solidity below 1 at the blade root means that the optimal chord length is not met towards the root of blade designs for high inclination angles and low tip speed ratios. It should be noted that in most Horizontal axis wind turbine designs, the blade design towards the blade root is dominated by structural considerations, and that this area would likely not be subject to aerodynamic optimisation for a realistic rotor design.





### 3.3 Survey of design space (optimised blade)

Figure 7 shows the power product, the corresponding thrust product and blade volume, each normalised by the respective value for an optimised H-Rotor, indicated by the blue square. The red diamond marks the location of maximum power product. It is clear that the limits on maximum chord length imposed by equation 2, lead to sub-optimal blade design at low tip speed ratios and high inclination angles. This can be seen as the blade volume contours begin to appear parallel with the x-axis, as the blade chord cannot be further increased.

As previously discussed, the blade volume can be taken as a proxy for blade mass and therefore cost. It is therefore interesting to understand the maximum power product achievable by blades of equal volume, to approximate the power gains that can be achieved without increase blade cost. This can be readily obtained by following blade volume contour lines equal to 1 from figure 7. Figure 8 (a) shows the power product for an optimised blade of equal volume to the optimal H-Rotor blade as a function of rotor inclination angle for a range of $l_r$. This is presented alongside the relative rotor area, again normalised by the optimal H-Rotor. Figure 8 (b) shows the optimal tip speed ratio at which the rotor operates. As is clear from the blade volume contours in figure 7, the tip speed ratio range considered for this study is not sufficiently wide to obtain optimal blade designs for combinations of large inclination angles and large $l_r$ values. For rotor geometries where the normalised volume of the optimal blade exceeds unity, the trace is extended in red, and the power product for the blade optimised at a tip speed ratio of 6 is shown.

Using an identical approach to that applied to un-tapered blades presented in section 3.1, the blade volume and rotor torque for rotor designs with an equal rated power are compared in figure 9. Contours showing the maximum bending stress, normalised by the value for the optimal H-rotor design are shown in black. Unlike the case for un-tapered rotors, it is clear that solutions exist within the optimised space that conform to the limitation on bending stress whilst decreasing blade volume and rotor torque. Additionally, for each of the cases, if a material innovation that facilitates higher bending stresses is applied, inclined rotor solutions also exist that lower both the blade volume and rotor torque compared to an equivalent H-rotor for all the aspect ratios considered here.

Figure 10 shows the minimum blade volume and rotor torque, normalised by the optimised H-Rotor design, for blade designs constrained by the maximum bending stress of the reference H-rotor as a function of rotor inclination angle. From figure 9, it is clear that for $l_r \geq 3$, the maximum bending stress for high inclination angles is below that of the reference H-rotor. For this reason, traces at which the maximum bending stress is lower than the constraint are represented by dashed, rather than solid, lines. For these values, more optimal blade designs are likely to exist at design tip speed ratios above those considered within this study. These lines are not smooth, primarily due to the coarseness of the original simulation grid in both $\lambda'$ and $\psi$, detailed in table 1.

It is clear that, for all aspect ratios considered here, a decrease in both blade volume and generator torque is achievable. As the blade length increases the potential for reductions in blade volume and rotor torque increase, with a maximum saving of 42% in blade volume for $l_r = 4, \psi = 20°$. A similar trend is shown in the potnetial reduction in rotor torque, although lower values

WIND
ENERGY
SCIENCE
DISCUSSIONS

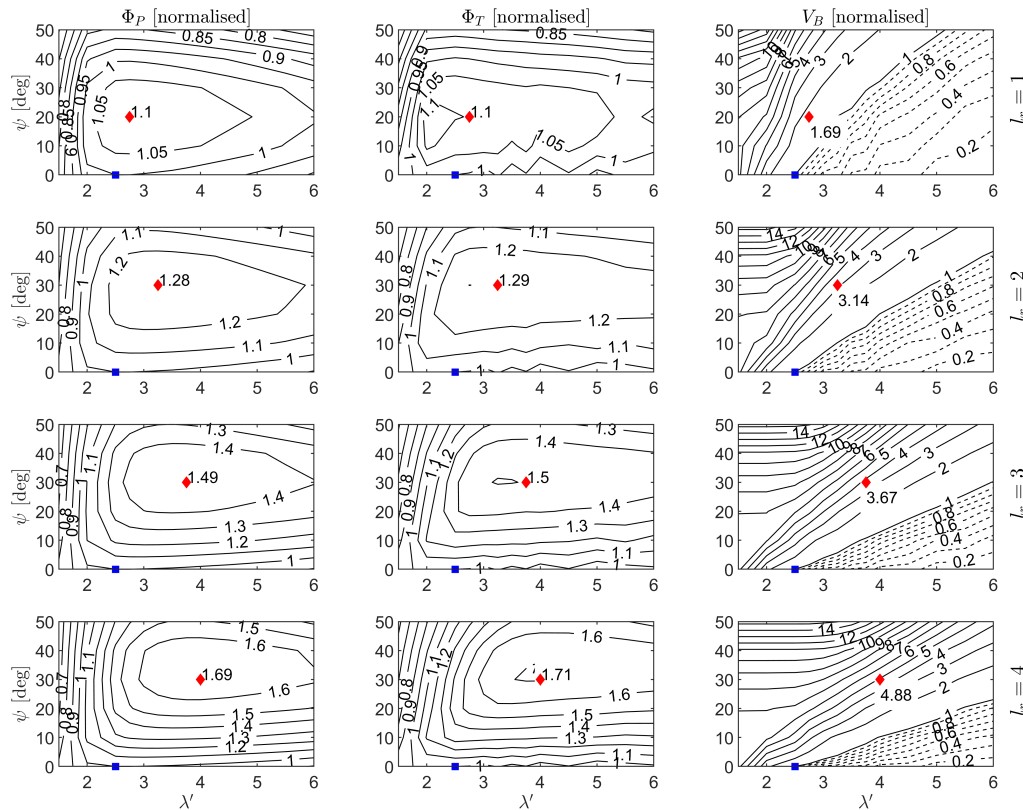

**Figure 7.** Power product, thrust product and blade volume, normalised by the optimal H-rotor value, presented as a function of design tip speed ratio and inclination angle for blade lengths $l_r = [1, 2, 3, 4]$. The optimal H-rotor design point is marked with a blue square, the optimal design for maximum power is marked with a red diamond.

are achievable, up to 9% for $l_r = 4, \psi = 20°$. It appears that there is correlation between the trends in blade volume and rotor torque, and it is clear that rotor designs exist which significantly reduce both blade volume and rotor torque simultaneously.

## 4 Discussion

### 4.1 Un-tapered blade performance

Section 3.1 presents the aerodynamic characteristics of a range of rotor designs with rotor aspect ratio, blade chord, and rotor inclination angle acting as the free variables. Figure 3 demonstrates that significant power gains can be achieved through introducing an inclination angle to untapered blades with increases of 10%, 28%, 48% and 68% available compared to the

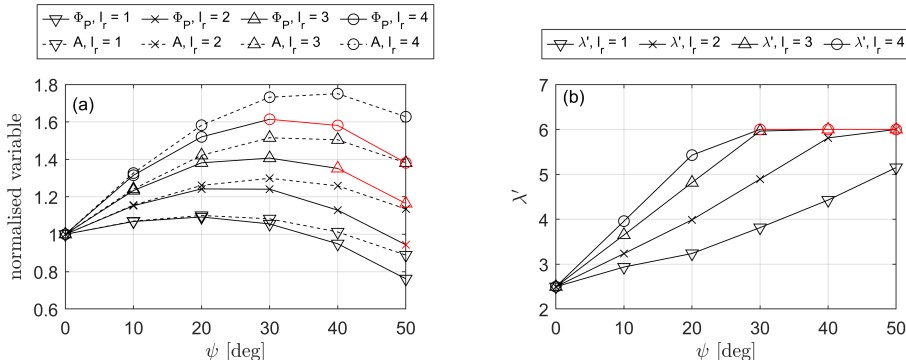

**Figure 8.** (a) Aerodynamic power and rotor area for an optimised blade with the same blade volume as the optimal H-rotor, normalised by the optimal H-rotor value, presented as a function of inclination angle. (b) corresponding optimal tip speed ratio. Data where trends indicate the optimal tip speed ratio is beyond the tip speed ratio limit are highlighted in red.

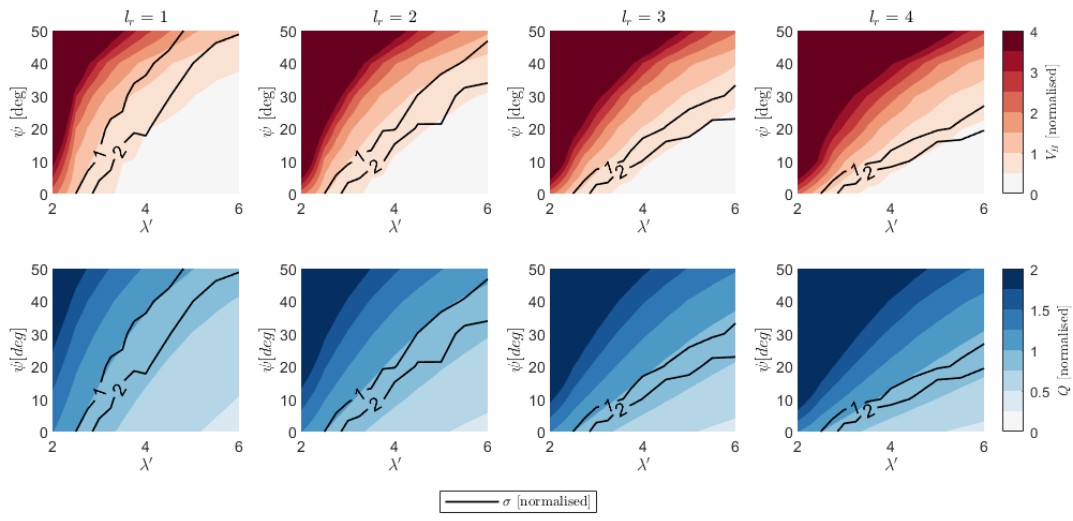

**Figure 9.** Blade volume and rotor torque contours, normalised by the optimal H-Rotor value, as a function of inclination angle, design tip speed ratio, and blade length. Contours of constant blade stress are plotted alongside in black.

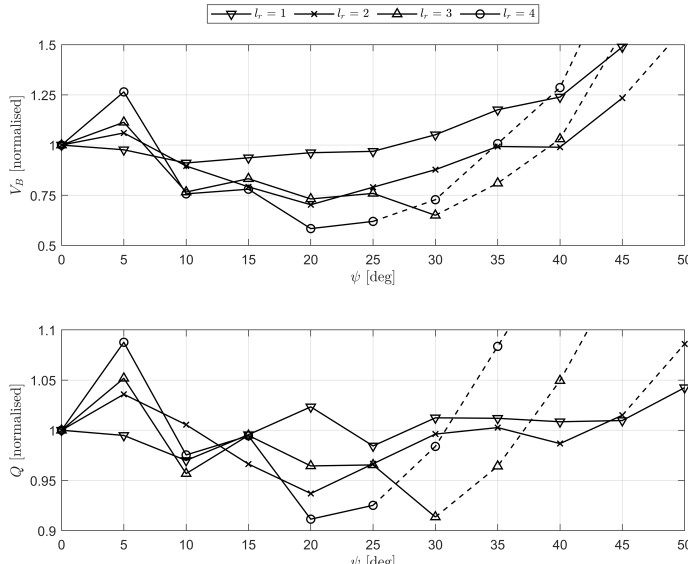

**Figure 10.** Minimum blade volume and rotor torque , normalised by the optimal H-rotor value, for rotor designs with constrained bending stress, as a function of rotor inclination angle. Dashed lines represent rotor configurations where the bending stress is below the critical limit.

optimal H-rotor design for blade lengths of $l/r_0 = [1, 2, 3, 4]$ respectively. These power gains are obtained through an increase in rotor swept area, and are therefore accompanied by an increase in thrust on the rotor by a similar value. The rotor designs which maximise the power available occur at a significantly higher chord length, and therefore do not represent a design solution that would decrease the relative cost of a blade compared to the optimal H-rotor, as is shown in figure 5.

For a blade of fixed chord length, figure 4 shows that inclining the blade increases the optimal tip speed ratio, and an
approximately linear relationship is observed with respect to the rotor inclination angle up to $50°$. Inclining the optimum H-rotor blade, without modifying the blade chord, has been shown to facilitate power gains 9%, 25%, 43%, and 60% for blade lengths of $l/r_0 = [1, 2, 3, 4]$. The blade volume and rotor torque for rotors with an equal power rating are shown in figure 5 and demonstrate that reductions in blade volume and rotor torque are achievable through inclining un-tapered rotor blades. However, when compared to contours showing the normalised maximum operational bending stress at the blade root, it is clear
that the most effective means of blade volume reduction without exceeding limits on the maximum operational bending stress is through reducing the blade chord for an H-rotor design, rather than introducing an inclination angle. This indicates that, if a constant chord distribution is employed, inclined rotor blades do not present an obvious case for cost of energy reduction. It should be noted, however, that as the inclined rotor blades will be smaller in plan-form area they will be subject to lower extreme loads in the parked case, and a more complete analysis is required to draw definitive conclusions in the case where
extreme, rather than operational loads are the design drivers.





## 4.2 Optimal blade design

A simple method of generating an optimal blade plan-form design is introduced in section 2.3, and is applied in section 3.2. It is clear that a taper is introduced to the blade as the inclination angle increases. The effect of the root and tip losses are shown to decrease the optimal blade chord at the blade root and tip and, aside from this effect at the blade root, the taper is monotonic. At low tip speed ratios and high inclination angles, the local blade chord towards the blade root is limited by the solidity condition imposed by equation 2.

## 4.3 Optimal blade performance

Section 3.3 presents the aerodynamic characteristics of the rotors employing the optimal blades described in section 3.2. Figure 7 demonstraits power gains of 10%, 28%, 49%, and 69% for blade lengths of $l/r_0 = [1, 2, 3, 4]$ are available compared to the optimal H-rotor design. In the same manor as the un-tapered case, increases in power are matched by a equivalent increase in the thrust on the rotor. Figure 7 also demonstrates the significant increase in blade volume accompanied by increasing inclination angle and decreasing design tip speed ratio. The power gain available to rotors with an equal blade volume to the optimal H-rotor is shown in figure 8, with values 9%, 24%, 40%, and 52% for blade lengths of $l/r_0 = [1, 2, 3, 4]$ shown to be achievable.

Figure 9 displays the change in blade volume and rotor torque for rotors with an identical power rating, demonstrating that increasing the design tip speed ratio decreases both blade volume and rotor torque, and that introducing an inclination angle increases both blade volume and rotor torque. However, when considering limitations on the maximum allowable bending stress, it becomes clear that V-rotors can reduce both the blade volume and rotor torque, compared to H-rotors under the same limitations on the maximum operational bending stress. This is further exemplified in figure 10, where the changes in blade volume and rotor torque are shown as a function of rotor inclination angle in the case where the maximum bending stress at the blade root is limited to the value of the optimal H-rotor design. Blade volume reductions of 9%, 30%, 35%, and 42% with corresponding rotor torque reductions of 3%, 6%, 9%, and 9% for blade lengths of $l/r_0 = [1, 2, 3, 4]$ are shown to be achievable.

These results apply a structural limitation to an aerodynamically optimised design. This serves to indicate if potential solutions exist within the aerodynamically optimised space that conform to the imposed structural limitation, and is not equivalent to any kind of full aero-structural optimisation. A more complete aero-structural optimisation is required to define any meaningful blade design. However, these results highlight the fact that the natural taper introduced through a purely aerodynamic optimisation can produce blade designs that conform to basic structural limitations whilst lowering the blade volume and rated torque of the rotor for a given power rating. This finding indicates that V-rotors have the potential to reduce the cost of energy compared to H-rotor designs.

In the case of horizontal axis wind turbines, thicker aerofoil sections are typically used at the blade root to increase the second moment of area and decrease the bending stresses. Although the thicker aerofoil sections lower the local rotor power coefficient, the local power coefficient of these blade sections contribute little to the overall rotor power coefficient due to the relative swept area. V-rotors introduce a similar trend, where the swept area at the blade root is significantly lower than the





blade tip, which is not the case in H-rotors. This can facilitate a similar approach to HAWT blade designs, and any future
aero-structural work on V-rotor blades should include multiple aerofoil sections in order to explore this possibility.

Additionally, a key benefit identified for straight bladed VAWT rotors is the ease of blade manufacture compared to HAWT
blades. In order to maintain this benefit, a simplified chord distribution should be introduced, employing a simple linear taper,
rather than an aerodynamically optimised design.

## 5    Conclusion

This paper presents a study focusing on the effect of blade inclination angle on the aerodynamic behaviour of V-rotors, under-
taken using a 2DAC aerodynamic model. The first section of the results focus on the effect of rotor inclination angle, blade
length (rotor aspect ratio) and blade chord length on the aerodynamic characteristics of VAWTs with un-tapered blades. It is
demonstrated that, although significant power gains are available through using inclined rotor blades, when considering lim-
itations on the maximum bending stress experienced by the blade, H-rotors remain the most economical option, allowing for
the minimum possible blade volume and rotor torque for any given power rating.

Following this, a simple procedure for optimising the blade chord distribution for maximum power coefficient is described
and implemented. It is shown that a taper is introduced into the blade through the aerodynamic optimisation, which has potential
structural benefits.

Finally, the aerodynamic characteristics of rotors employing these optimised blades are presented. It is shown that significant
increases in rotor power are available through the utilisation of optimised inclined blades, however designs at low tip speed
ratio have significantly larger blade volumes and therefore considered un-economical. The comparison of blade volume and
rotor torque for rotors of equal power utilising optimal blades demonstrates the potential for cost of energy reductions using
V-rotors, and the natural taper introduced through optimal aerodynamic design reduces the blade root bending stress and
facilitates designs that significantly reduce both the blade volume and rotor torque. Implying potential blade cost reductions of
9-52% and drivetrain cost reductions of 2-9% compared to an aerodynamically optimised H-rotor.

The higher optimal tip speed ratio of rotor designs utilising inclined blades is also a potential benefit for wind turbine
concepts such as the X-Rotor, where higher primary rotor tip speed ratios are desirable to ensure efficient power conversion
using secondary rotors Morgan et al. (2024).

Future work must consider a more detailed aero-structural design, considering both fatigue and extreme loading, and in-
cluding inertial and gravitational loads on the blade. Additionally, the effects of rotor loads on the support structure should be
considered.

*Data availability.*    The aerodynamic simulation data is available on request from the corresponding author.





## Appendix A: Kinetic flux through VAWT rotor geometries

This appendix discusses the effect of wind shear on the kinetic flux through a straight bladed VAWT rotor.

The area swept by a VAWT is given by

$$A = 2 \int_{z_{min}}^{z_{max}} r(z) d(z) \tag{A1}$$

where z represents the vertical coordinate, and r represents the rotor radius. In plane inflow, this rotor area can be used as a proxy for the power available to the turbine, however, due to the wind shear profile, the kinetic flux is non-uniform over the rotor height and should be included in this calculation. The wind shear profile can be approximated using the power law:

$$U(z) = U_0 \left( \frac{z}{z_0} \right)^{\alpha} \tag{A2}$$

where $U_0$ refers to the reference wind speed, $z_0$ represents the sample height, and the shear exponent $\alpha$ is dependent on the environmental conditions, but can be assumed to be 0.12 for a typical offshore site. The kinetic flux is therefore given by

$$I(z) = \frac{1}{2}\rho U(z)^3 = \frac{1}{2}\rho \left( \frac{U_0}{z_0^{\alpha}} \right)^3 z^{3\alpha}. \tag{A3}$$

where $\rho$ represents the rotor area. The power available to the turbine is therefore given by

$$P = \frac{1}{2}\rho U_0^3 \left[ 2 \int_{z_{min}}^{z_{max}} r(z) \left( \frac{z}{z_0} \right)^{3\alpha} d(z) \right] = \frac{1}{2}\rho U_0^3 \tilde{A}. \tag{A4}$$

where $\tilde{A}$ can be considered the effective swept area of the turbine, weighted by the power available due to the wind shear profile. The rotor geometry can be described using the variables in figure 2, and length scales can be non-dimensionalised by the inner radius of the rotor. In addition to this, the rotor's location within the shear profile can be related to length scale of the rotor through minimum height $z_{min}$, and the sample height $z_0$. Here it is considered that the minimum rotor height is

approximately equal to the inner radius, $z_{min}/r_0 = 1$, this is equivalent to a 2.5MW H-Rotor with an aspect ratio of 2 having a tip clearance equal to 25m. The location of the sample height $z_0$ is considered to be equal the center of the equivalent H-rotor i.e. $z_0/r_0 = (1 + l_r)/2$. This allows $A$ and $\tilde{A}$ to be presented non-dimensionally, as shown in figure A1 (a). Comparing the effective area with the geometrical area compares the the maximum power available to a rotor of a given geometry in a realistic boundary layer to the same rotor in plane inflow, with velocity equal to the reference velocity. The power available to the rotor

peaks at a lower inclination angle than that which maximises the geometrical area, and the peak effective area is lower than the peak geometrical area for each aspect ratio, with a relative decrement of 0.6% for $l_r = 1$ increasing up 2.2% for $l_r = 5$.

The relative difference between the effective area and the geometrical area, is equivalent to the difference between the maximum power available to turbine in sheared vs plane inflow, this is shown in figure A1 (b).

It can be seen that the inclusion of the effects of wind shear will decrease the expected power output for most rotor designs,

with a small ($< 0.5\%$) increase in the power production being expected for high aspect ratio rotors with an inclination angle



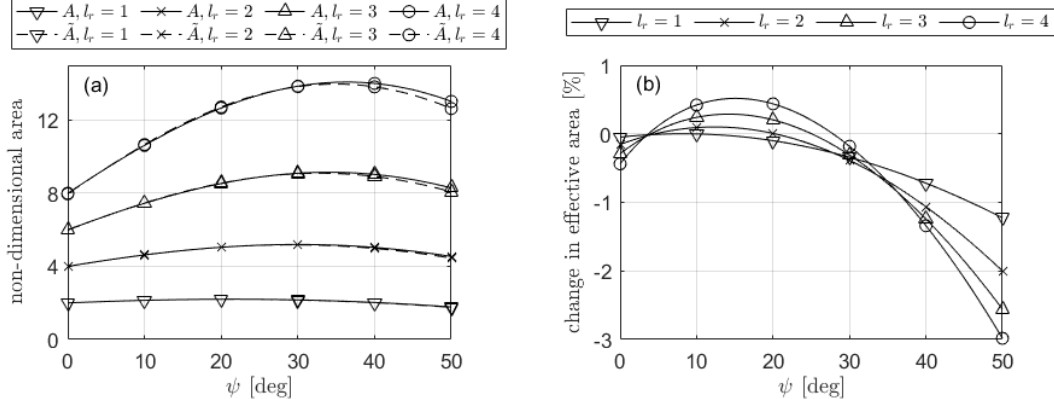

**Figure A1.** (a) Rotor area, and effective rotor area non-dimensionalised by the rotor inner radius presented as a function of rotor inclination angle for various blade lengths. (b) Relative difference between effective and geometrical rotor area presented as a function of rotor inclination angle for various blade lengths.

between 10 and 25 degrees. The decrement in power available increases significantly as the inclination angle increases, up to a 3% decrement at an inclination angle of $50°$. The numerical values generated by these results are, of course, dependent on the rotor's location within the boundary layer. As the rotor moves higher into the boundary layer, the relative change in velocity over the height of the rotor decreases, decreasing the effects. However the large relative performance difference, and the non-uniformity of this performance difference across both effective aspect ratio and inclination angle demonstrates the need to include wind shear effects in further evaluation of the efficacy of VAWT rotor geometries. The inclusion of wind shear effects is further complicated by the fact that the rotor power coefficient in non-uniform over the rotor height, and that with a rotor designed for plane inflow, blade sections will be not be operating at their design tip speed ratios. A full study of these effects is outwith the scope of this paper, however the requirement for their inclusion in more precise design work has hopefully been demonstrated in this appendix.





*Author contributions.* L.M.: Study conceptualisation, simulation work, data analysis, manuscript preparation. A.K.A.: Manuscript preparation and technical review. W.L.: Study conceptualisation, manuscript preparation. J.C.: Manuscript preparation.

*Competing interests.* No competing interests are present

*Acknowledgements.* The authors would like to acknowledge Adhyanth Giri Ajay and Professor Carlos Ferreira for the development and
dissemination of the 2DAC aerodynamic code as part of the X-Rotor project, and for insight on VAWT aerodynamics. The authors would like to acknowledge their funding grants: XROTOR, EU H2020, Grant/Award Number: 101007135; Centre for Doctoral Training in Wind and Marine Energy Systems, EPSRC, Grant/Award Number: EP/S023801/1





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
