# Peer review of "Effect of Blade Inclination Angle for Straight Bladed Vertical Axis Wind Turbines"

_Wind Energy Science, 2024_

## Referee Comment (RC2)

[referee-annotated manuscript omitted]

---

## Author Comment (AC1)

**Review Responses**

June 2024

The authors would like to thank the reviewers for their comments on the paper. We believe that your contribution has helped improve the paper. Each reviewer is responded to in the corresponding section of this response and each numbered comment is responded to.

**Reviewer 1**

1. "Given the solidities and tip-speed ratios considered for the analysis, ignoring the flow curvature and unsteady aerodynamics/dynamic stall effects might bias the obtained trends, especially in terms of power coefficient and maximum blade stresses. Therefore, the limitations of the adopted methodology should be assessed against simulations performed with a higher-fidelity method, at least for a couple of points at the extremes of the chosen investigation domain"

   The 2DAC code has been tested against high fidelity codes [3], the results from this showed a 1.8% over estimate in the power coefficient and a 16% over estimate in the upper blade root bending stress moment and 2.3% increase in the maximum lower blade bending stress compared to the blade resolved 3D URANS CFD results. We hope that this verification of the code against blade resolved URANS results, alongside the further verification against vortex models and other momentum models presented in [3](reproduced in figure 1, showing the integrated forces on the upper and lower blades of the X-Rotor) are adequate to demonstrate that the modelling method is valid for the purposes of this study. This discussion has been added to the text (see lines 198-203).

2. The Prandtl tip loss factor has been conceived for HAWTs with straight blades. How do you justify its application to a VAWT with a largely "three-dimensional" geometry as the one considered here?

   The authors agree that the Prandtl tip loss factor was not designed for VAWTs, and agree that further research work is required to investigate VAWT specific tip loss model. However, we would justify our approach on the grounds that Prandtl tip loss factor has been successfully applied in momentum based VAWT models in multiple instances.

Additionally, noting that Prandtls tip loss factor is derived from the simplification of a helical wake of trailing vorticity from a HAWT into a series on impenetrable disks, Sharpe [2] proposed that the vortex sheets in the wake of an H-rotor could be represented in a similar manner with the sheets being shed twice per blade per revolution and employed a Prandtl type tip correction factor. Paraschiviou also used a similar formulation [6]. Prandtl type tip loss models have been shown to reproduce the results of mid fidelity vortex methods for V rotors in terms of revolution averaged coefficients and blade root loads when applied to DMS models [4]. Additionally, the specific tip loss formulation has been verified against vortex models for multiple operational conditions and blade resolved CFD for a single operating condition in [3]. Paraschiviou's book is now referenced in the text (see line 156).

3. The use of a single set of polar data is reasonable, given the large blade Reynolds number involved. Have you verified, nonetheless, the accuracy of the obtained dataset, especially in the post-stall region?

The accuracy of the post stall region has been shown to be important [1], and the post stall data used in these polars has not been directly verified though simulation or experimentation. A reference discussion this issue has been added to the discussion in the dynamic stall section of the limitations (lines 175-177). The authors would again point to verification of the code in reference to the X-Rotor turbine as presented in [3] which has demonstrated that the current polars (identical to those used in the referenced study) are sufficient for this investigation.

4. Abstract is too focused on the results and lacks the methodology part. Please re-write it to balance it out;

The abstract has been modified to better describe the methodology used (lines 6-9).

5. Section 2: it is not clear how a 2D tool is applied to the simulation of a 3D VAWT rotor, especially in presence of inclined blades. Please integrate this section to clarify this aspect;

The text has been modified to include a more in depth description of the application of the 2DAC model to 3D turbines (lines 147-154) .

6. Line 194: how is the $\lambda'$ variable defined?

the tip speed ratio has now been defined (see line 231)

7. Line 236: how was the optimal H-rotor designed? Please clarify

The text has been modified to clarify that the optimal H-rotor is defined as the H-Rotor with the chord length that maximises the power coefficient (see line 275 - 276)

[Figure]

Figure 1: Caption

**Reviewer 2**

1. Explain why high tip speed ratios increase applicability to rotors using secondary rotors

   The text has been modified to state explicitly that the high tip speed ratios are required for efficient power conversion between primary and secondary rotors (see line 19-20). The study of this effect is detailed in [5] which is cited in the text.

2. Elaborate or re-word on VAWT wakes (re-energises) (see line 29) The word recover rather than re-energise has been used

3. Which VAWT design concepts The text has been modified to highlight that a variety of VAWT design concepts are receiving renewed interest (see line 31).

4. Please explain how V-rotor blades operate at variable tip speed ratios The text has been modified to clarify that V-rotor blade sections operate at different tip speed ratios due to the change in radial coordinate along the blade (see line 90).

5. "completed to" The phrase "completed to" has been replaced with the phase, "The study is performed with the goal of understanding" (see line

117).

6. Modeling only has one l All uses of the word modeling have been edited to correct the spelling

7. out-with to outside outwith has been replaced with outside (see line 172)

8. why? What is the justification for this variation? This variation is justified by the hard limit on solidity ($\sigma \leq 1$. Since the solidity is a function of the chord length and the inclination angle, the maximum chord length is a function of the inclination angle.

9. define TSR The tip speed ratio is now defined (see line 231)

10. Performed rather than completed The word completed has been replaced by the word performed (see line 237)

11. Why such a high reynolds number? The Reynolds number is consistent with multi-MW offshore VAWTs as per the annotated comment supplied on the review.

12. replace following this with next The phrase "following this" has been replaced with "next"

13. This scaling does not preserve the whole concept of similitude as grounds for comparison. Can you please explain how this scaling is better than the traditional non-dimensional groups for the comparison of performance An additional section has been added in the introduction (lines 96 -115) which highlights specifically the effects of changing the blade inclination angle. Changing the blade inclination angle increases the swept area of the rotor whilst using the same amount of material (blade and crossarm length), since the two rotors with a different inclination angles are not geometrically similar, using the traditional pi group variables $C_P$ and $C_T$, do not capture the effects of blade inclination, hence the introduction of $\Phi_P$ and $\Phi_T$.

14. This needs more clarification. What is the equivalent H-Rotor? How PT then allow comparison of loads if the choice of A is different/ Traditionally CP and CT would be normalised for same reference swept area when comparing two different turbines with different swept area.

   The definition of the equivalent H-Rotor (an H rotor with the same value of $l_r$) has been added to the text (see line 270). The significant increase in rotor area when blades are inclined (up to 1.8 times) means that using the H-Rotor area as a reference area for $C_P$ and $C_T$ calculation as this would yield $C_P$ values which would exceed the Betz limit, due to the difference in swept area, and this would be an implausible and improper result.

15. You should be able to get the same results for CP and CT by normalising using same swept area as that of a H-Rotor! See above comment

16. No this is not correct unless proven/ Both high solidities and TSR have been observed to produce less power and reduce CP and CT

    The results clearly show that, for high inlination angles, the trend in optimum tip speed ratio goes beyond the range of tip speed ratios considered in this study for high aspect ratio and high inclination angle designs (see figure 4 page 13).

17. Please explain the effect of each variable in the discussion Whilst the results are presented in section 3, section 4 includes a thorough discussion of the effect of each variable.

18. Again these Fig results are not clear and perhaps misleading e.g. comparing points lr =3 is a larger rotor than lr = 2 normalising with psi - will of course lead to large phi P and phi T by intuition. But it does not convey the advantage of one psi over the other unless areas are the same so you need to show results for the same scaled swept areas to see advantage

    The effect of blade inclination is to increase the swept area, and the increase in swept area as a function of blade inclination angle increases as the aspect ratio increases. It is not misleading to present the fact that the larger area increase available to high aspect ratio rotors leads to larger potential increase in power production. Through normalising each aspect ratio to the results for the $\Psi_P$ value at that aspect ratio at $\psi = 0$, this figure demonstrates how the rotor aspect aspect ratio changes both the optimal $\psi$ value and the available increase is power capture for a rotor with fixed geometrical parameters (blade length and cross arm length).

19. Only in terms of rotor size, better to show the effect for the same swept area rotor designs! The effect of increase in rotor size (balanced by the decrease in rotor efficiency) is what we are trying to capture in this study. The text of the introduction and the discussion of $\phi_P$ and $\phi_T$ has been modified to try to clarify that this is the case.

**References**

[1] Alessandro Bianchini, Francesco Balduzzi, John M Rainbird, Joaquim Peiro, J Michael R Graham, Giovanni Ferrara, and Lorenzo Ferrari. An experimental and numerical assessment of airfoil polars for use in darrieus wind turbines—part ii: Post-stall data extrapolation methods. *Journal of Engineering for Gas Turbines and Power*, 138(3):032603, 2016.

[2] L.L. Ferris and D.J. Sharpe. *Wind Energy Conversion Systems, Chapter 4*. 1990.

[3] Adhyanth Giri Ajay, Laurence Morgan, Yan Wu, David Bretos, Aurelio Cascales, Oscar Pires, and Carlos Ferreira. Aerodynamic model comparison for an x-shaped vertical-axis wind turbine. *Wind Energy Science Discussions*, 2023:1–25, 2023.

[4] Laurence Morgan and William Leithead. Aerodynamic modelling of a novel vertical axis wind turbine concept. *Journal of Physics: Conference Series*, 2257, 5 2022.

[5] Laurence Morgan, William Leithead, and James Carroll. On the use of secondary rotors for vertical axis wind turbine power take-off. *Wind Energy*, 2024.

[6] Ion Paraschivoiu. *Wind Turbine Design: With Emphasis on Darrieus Concept.* 2002.

---

## Author Response (AR2)

**Review Responses**

September 2024

I would like to thank reviewer 3 for their comments on the paper. The comments on both the modelling, validation and the presentation of the results, has helped improve the paper, and improve myself as a researcher.

**Reviewer 3**

1. "I would recommend to implement at least: - the flow-curvature correction, which is well documented in the literature of VAWT (the Gaude's model in particular is of straightforward implementation, see Dyachuck and Goude, 2015, Energies);"

   The 2DAC code has been modified to include the curvature correction described in (Dyachuck and Goude, 2015, Energies). This correction was found to produce a deviation in both power coefficient calculation and blade force results, this is now presented in Appendix 2.

2. "I would recommend to implement at least: a modeling of the drag induced by the struts, whose weight is expected to change significantly depending on the inclination of the blades (by referring to the wide literature on the topic, see for example the paper of Bianchini et al., 2017, Energy Conversion and Management)."

   The correction for the parasitic drag on the rotor crossarm has now been modelled using the correction presented in (Bianchini et al., 2017, Energy Conversion and Management). Whilst the headline results have not changed significantly, the effect of including the strut drag has been seen to increase the optimum solidity and decrease the optimum tip speed ratio for both the un-tapered, and the optimised blades. The strut correction has the largest effect for lower rotor aspect ratios.

3. "With these added models the authors should repeat the validation performed in the previous paper they cite, and introduce in this paper a brand new section reporting these new validation results."

   A new subsection concerning model validation has been included where the power coefficient and integrated blade forces for the X-Rotor upper and lower blades, as calculated using a blade resolved URANS simulation,

are compared the 2DAC model used in this study. In addition, the effect of the proposed curvature correction is presented in appendix 2.

4. "As a second main consideration, the authors propose to use as metric for the optimization the parameters PhiC and PhiT, which are dimensional quantities combining the power and thrust coefficients with the swept area. My impression is that these parameters are of doubtful relevance"

$\Phi_P$ and $\Phi_T$ are no longer presented or introduced, and the change in rotor area is presented seperately to the change in rotor power coefficient. In order to understand the coupled effect, the normalised power and thrust are presented alongside the change in power coefficient.

---

## Author Response (AR3)

**Review Responses**

October 2024

I would like to thank all of the reviewers for their work reviewing the paper. Their comments have improved the paper, and have provided guidance for future research. I would also like to thank the associate editor and chief editor for considering the paper and approving it for publication.